# Exploring the Mystery of the Sweetness of Baijiu by Sensory Evaluation, Compositional Analysis and Multivariate Data Analysis

**DOI:** 10.3390/foods10112843

**Published:** 2021-11-17

**Authors:** Yulu Sun, Yue Ma, Shuang Chen, Yan Xu, Ke Tang

**Affiliations:** 1Lab of Brewing Microbiology and Applied Enzymology, Jiangnan University, 1800 Lihu Avenue, Wuxi 214122, China; sunyulu1120@163.com (Y.S.); 7160201007@vip.jiangnan.edu.cn (Y.M.); chengshuang1984@163.com (S.C.); biosean@126.com (Y.X.); 2Key Laboratory of Industrial Biotechnology of Ministry of Education, School of Biotechnology, Jiangnan University, 1800 Lihu Avenue, Wuxi 214122, China; 3State Key Laboratory of Food Science and Technology, Jiangnan University, 1800 Lihu Avenue, Wuxi 214122, China

**Keywords:** sweetness, baijiu, volatile compound, sensory-guided fractionation, multivariate data analysis

## Abstract

Sweetness is an important Baijiu quality marker, but there is limited research on it. In this study, the main contributors to Baijiu sweetness were identified by “sensomics” combined with “flavoromics”. A total of 43 volatile compounds (mostly esters) were found that appeared to contribute to Baijiu sweetness, through sensory-guided fractionation and compositional analysis. Correlation analysis between the volatile composition and perceived sweetness of 18 Baijiu samples with different sweet intensities identified 14 potential contributors. Additional testing verified that combining the 14 compounds reproduced Baijiu sweetness exactly, and omission testing identified ethyl hexanoate, hexyl hexanoate and ethyl 3-methylbutanoate as the major contributors to Baijiu sweetness. These findings not only broadened our understanding of Baijiu sweetness, but also highlighted the major contribution of volatile compounds to sweetness perception, knowledge which may facilitate future flavor modification of a wide variety of foods and beverages.

## 1. Introduction

Chinese Baijiu is one of the major distilled liquors in the world, alongside brandy, whiskey, vodka, rum, gin, etc. [1]. Nearly 7.5 billion liters of Baijiu were produced, with sales revenue of USD 90.33 billion in 2020 [2]. Compared with other distilled liquors, the production of Baijiu is a unique and complex process, which includes various raw materials (grains: sorghum, rice, wheat, or millet and Qu: grain-based starter culture); solid-state fermentation (saccharification and spontaneous fermentation under mixed-culture conditions); distillation (directly from the solid-state fermentation); and aging (in pottery jars) [3]. This special manufacturing process produces the characteristic rich flavor of Baijiu [4]. Taste (sourness, bitterness, saltiness, sweetness) is an important factor determining Baijiu quality, with sweetness being the main quality determinant. In the past decade, research on Baijiu flavor has mainly focused on analysis of the many aroma compounds [4,5], bitter and astringent compounds [5], and quantitative prediction of pungency [6]; to the best of our knowledge, no studies have been performed on the sweetness of Baijiu.

Sweetness is perceived when a target molecule binds to a sweetness receptor (T1R3 and T1R2) [7]. Compounds such as sugar, synthetic and natural sweeteners, and sweet proteins, or amino acids are all capable of activating sweetness receptors [8,9]. Baijiu is a distilled liquor with relatively few non-volatile compounds, although some non-volatile sweet polyhydroxy compounds (glucose, glycerol, fructose, sucrose etc.) have been identified, the concentrations of these substances in Baijiu are far below the threshold of sweetness perception [10]. Ethanol has been reported to have perceived sweetness and presumably can activate sweetness receptors [11]; however, the sweetness of different types of Baijiu with the same ethanol content varies widely. Therefore, there must be other yet to be discovered compounds that contribute to Baijiu sweetness.

“Sensomics” is a method to identify the key flavor compounds in foods, in which the food matrix is divided into several fractions and the key compounds are isolated and identified by sensory-guided fractionation [12,13]. However, the practical application of sensory-guided fractionation techniques has some limitations that the preparative fractionation cannot always resolve at the single compound level. In that case, “flavoromics” may be an extremely useful tool to assist in locating the key flavor compounds in food products by applying multivariate data analysis (MVDA) to investigate the correlation between the chemical profiles and sensory evaluation of samples [14]. Many key flavor compounds in food have been identified in this way, such as the retronasal “burnt” flavor in soy sauce aroma type Baijiu [15], “stone fruit” flavor in wine [16], and bitter modulators in brew [17].

Accordingly, the aim of this study was to characterize the main contributors to the sweetness of Baijiu by using sensomics, assisted by flavoromics. The aims of this analysis were: (i) to identify compounds that appear to contribute to the sweetness of Baijiu, by using sensory-guided fractionation and compositional analysis; (ii) to assess the relative contribution of the compounds responsible for the differences in Baijiu sweetness, through MVDA; and (iii) to verify the key compounds responsible for the sweetness of Baijiu via sensory validation. The findings obtained will aid understanding of the chemical basis of Baijiu sweetness and identify the main contributing compounds.

## 2. Materials and Methods

### 2.1. Baijiu Samples

A total of 18 Baijiu samples (labeled as S1–S18) with different sweetness intensity were selected (Appendix A). All the samples were diluted with ultrapure water to the same ethanol content (50%, *v*/*v*) and were stored at 4 °C before analysis. Sample S1 was used for the sensory-guided fractionation/isolation.

### 2.2. Chemicals

All chemical standards and internal standards (IS) were of the highest available purity (GC-grade). Ethyl pentanoate (≥98%), ethyl hexanoate (≥99%), hexyl octanoate (≥98%), ethyl 3-methylbutanoate (≥99%), ethyl heptanoate (≥98%), isoamyl butanoate (≥98%), hexyl hexanoate (≥98%), isoamyl octanoate (≥97%), isobutyl hexanoate (≥98%), ethyl butanoate (≥98%), isoamyl hexanoate (≥98%), ethyl octanoate (≥98%), octanal (≥98%), and amyl hexanoate (≥98%) were from J&K Scientific (Beijing, China). 2,2-dimethylpropanoic acid (≥98%, IS1), L-Menthol (≥99%, IS2), 2-phenylethyl acetate-d3 (≥98%, IS3), 2-methoxy-d3-phenol (≥98%, IS4), amyl acetate (≥99%, IS5), and a mixture of C7–C30 alkanes were from Sigma-Aldrich (Shanghai, China). Ethanol [high-performance liquid chromatography (HPLC) grade] was from J&K Scientific. Sodium chloride (NaCl) was purchased from China National Pharmaceutical Group Corp (Shanghai, China). Ultrapure water was obtained from a Milli-Q purification system (Millipore, Bedford, MA, USA).

The reagents used in the sensory section were all food grade. Ethyl pentanoate (≥98%), ethyl hexanoate (≥98%), hexyl octanoate (≥97%), ethyl 3-methylbutanoate (≥98%), ethyl heptanoate (≥98%), isoamyl butanoate (≥98%), hexyl hexanoate (≥97%), isoamyl octanoate (≥98%), isobutyl hexanoate (≥98%), ethyl butanoate (≥98%), isoamyl hexanoate (≥98%), ethyl octanoate (≥98%), octanal (≥95%), amyl hexanoate (≥98%), and food grade ethanol were from Sigma–Aldrich (Shanghai, China).

### 2.3. Sensory Analysis

The sensory evaluation in this study included descriptive sensory analysis of 18 Baijiu samples with different sweetness intensities, six fractions (volatile components, nonvolatile components, and 4 fractions distilled from different vacuums) distilled from sample S1 and 50% (*v*/*v*) ethanol aqueous solution; addition and omission tests of the compounds that were positively associated with Baijiu sweetness. The entire process of sensory evaluations from sensory panel recruiting to sample testing was shown in Appendix A.

#### 2.3.1. Recruiting and Training of Sensory Panel 

Panel candidates were recruited and selected form the students of Jiangnan University. They were asked to fill out a questionnaire about their basic information, willingness and interest to participate in the group, as well as health status (participants were excluded if they were smokers, wore dentures, had sinus problems, were on special diets, had specific food dislikes, had allergies/intolerances, had phenylketonuria, or were diabetic or pregnant). Priority was given to the candidates who had sensory analysis experience and are familiar with Baijiu. After the questionnaire, the candidates were required to complete sensory ability tests for aroma identification, perception and ranking of basic taste (including sour, sweet, bitter, salty and astringent) and response scales test in separate sensory booths. Fifty-five candidates were selected who had achieved at least 70% acuity. The general training (four sessions, 1 h/session, twice a week) mainly focused on recognition of different perceptions of sweetness and utilizing an intensity scale for rating sweetness using a 15-point unstructured line scale (0 = none, 15 = extreme; sliding bar scale on a computer), based on a series of sucrose solutions of different concentrations (0–20 g·L^−1^). Their performance was monitored to ensure the effectiveness of the training. Thirty-six healthy panelists (27 females and nine males, 18–25-years-old) were selected. All participants provided informed consent in line with the Helsinki Declaration for experimentation and were compensated for their participation in the study.

The sensory panel was conducted in two separate groups with specific objectives for each. Group I (12 members, nine female and three male) was involved in scoring the sweetness of the sample by descriptive sensory analysis. Group II (24 members, 18 female and six male) was involved in distinguishing and verifying the sweetness characteristics of the identified compounds, by two-alternative forced choice (2-AFC). Group I panelists attended seven Baijiu-specific sensory training sessions and one discussion session (1 h per session, twice a week) over 1 month. The Baijiu-specific section (7 sessions) familiarized the panelists with the sweetness characteristics of Baijiu samples. During these sessions, Baijiu samples with different sweetness levels were presented to the panelists, they were guided to rate the relative sweetness, using a 15-point unstructured line scale (0 = none, 15 = extreme; sliding bar scale on a computer). The panelists’ performance was assessed in terms of their ability in consistency, stability and repeatability for giving scores before sample evaluation by panel check [18]. During the last discussion section, panelists were asked to select three samples as external references for descriptive sensory analysis. The overall sweetness of the three reference samples was determined by consensus, resulting in their sweetness intensities being rated as 3, 8 and 12, respectively. Group II panelists attended one training session to become familiarized with the sweetness of Baijiu and the test method of 2-AFC.

#### 2.3.2. Pretreatment of Fractions 

Sample S1 was selected as the representative sample for sweetness fractionation screening by the sensory-guided strategy, because it was rated as the sweetest sample by all the panelists. To obtain different fractions for descriptive sensory analysis, sample S1 (500 mL) was fractionated using a rotary evaporator (Buchi Laboratories, Flawil, Switzerland) coupled with a low temperature circulating bath (DLSB-5/10, Gongyi Yuhua Instruments, Gongyi, China) and a water recycling vacuum pump (SHZ-DIII, Gongyi Yuhua Instruments, Gongyi, China). 

In the first step, the sample was separated into volatile and non-volatile components by rotary evaporation at 55 rpm/min and 40 °C, with the vacuum pump operating at 30 hPa. Both fractions were subjected to descriptive sensory analysis, to determine their sweetness, after adjusting their ethanol contents to 50% (*v*/*v*) with food-grade ethanol (the same ethanol content as Sample S1). Both the volatile and nonvolatile components were prepared the day before sensory evaluation and were stored at −20 °C until needed. 

In the second step, the volatile fraction from the first step was separated by rotary evaporation under four different vacuum settings. Volatile fraction (100 mL) in a 250 mL flask was vacuum distilled in the rotary evaporator at 55 rpm/min and 85 °C, to produce samples A–D, with the vacuum pump set at 800 (A), 600 (B), 400 (C) and 200 (D) hPa. Distillation was considered complete at each vacuum setting when no liquid drops had condensed for 5 min. The four fractions were then adjusted to the same ethanol content (50%, *v*/*v*) and were subjected to descriptive sensory analysis to determine their sweetness. The fractions were prepared the day before sensory evaluation and were stored at −20 °C until needed.

#### 2.3.3. Descriptive Sensory Analysis 

There were 25 samples involved in descriptive sensory analysis, including the 18 Baijiu samples and the six sample S1 fractions (volatile components, non-volatile components, and 4 fractions distilled from different vacuums) and one 50% (*v*/*v*) ethanol aqueous solution as black control. These samples were divided into three different sessions (around 8 samples for each session) and their distribution was counterbalanced for all panelists. All samples were evaluated 4 times in 4 different rounds which were conducted on 4 different days, so the total number of evaluations was 12 sessions (three sessions for a round). All the testing sessions took place in a specific room equipped with individual booths and air-conditioned at 20 °C. 

The overall sweetness of the samples was rated on a 15-point unstructured line scale (described above) by sensory Group I (12 members). To prevent cross-model interactions with olfactory cues, all panelists were required to wear nose clips before the test. Each testing session began with a “warm-up’, in which the panelists were required to taste the three reference samples (sweetness = 3, 8, 12), to calibrate their sweetness intensity scaling. After the warm-up, panelists were asked to taste 2 mL of each sample and rate the sweetness intensity within 10 s on the 15-point scale, then spit out the sample. Between samples, the panelists were asked to rinse their mouths with distilled water, to eat some plain crackers for 30 s and finally to rinse again with distilled water for another 45 s to minimize any carry-over effect from the previous sample. In each test session, samples were randomly presented to each panelist.

#### 2.3.4. Addition and Omission Tests 

Two-alternative forced choice (2-AFC) testing was performed by sensory Group II in addition and omission tests, to ascertain the sweetness contribution of 14 identified potential sweet compounds isolated from Baijiu. Samples for the addition test included Baijiu sample S1, blank control (50%, *v*/*v* ethanol aqueous solution) and Model S1 (MS1), which was prepared by adding the 14 compounds at the same concentration as in sample S1 (Sample after being diluted, 1.4 dilution factor) to 50 mL of aqueous ethanol (50%, *v*/*v*). The pair S1 and MS1 was evaluated first, to determine whether the 14 compounds could simulate the sweetness of Baijiu. The pair MS1 and blank control were evaluated to evaluate the sweetness difference between the model sample and the 50% (*v*/*v*) aqueous ethanol. Samples for the omission test were prepared by omitting each of the 14 compounds in turn, from the sample MS1 to determine the sweetness contribution of the omitted compound and each of these 14 samples was compared with MS1. The 16 sample pairs (two pairs for the addition test and 14 pairs for the omission test) were divided into two different sessions (8 pairs for each session) to be evaluated and their distribution was counterbalanced for all panelists. All the samples in the 2-AFC tests were blinded with randomized three-digit codes and the order of presentation of the two samples in a pair was randomized. To evaluate a sample pair, subjects were required to wear nose-clips and they were asked to taste 2 mL of the sample and to choose the sweeter one. The subjects were instructed to clear their mouths with crackers and water and to take a short break between sample pairs. 

### 2.4. Identification of Aroma Compounds 

#### 2.4.1. Gas Chromatography with Flame Ionization Detection (GC-FID)

As described previously [19], an Agilent (Santa Clara, CA, USA) 6890N gas chromatograph, fitted with a flame ionization detector, was employed to identify and quantify the major sweet compounds isolated (Appendix A). Baijiu samples (1 μL) with internal standard (IS5: amyl acetate, 174.24 mg·L^−1^) added were directly injected into the gas chromatograph in split mode (split ratio = 10:1). Nitrogen was used as the carrier gas at a constant flow rate of 2 mL/min, and a DB-Wax column (30 m × 0.25 mm i.d., 0.25 μm film thickness, J&W Scientific, Folsom, CA, USA) was used for separation. The column temperature was programmed as follows: initially 35 °C for 2 min; increased to 70 °C at 3.5 °C/min; increased to 180 °C at 5 °C/min; increased to 10 °C/min to 200 °C and held for 5 min. The injector and detector temperatures were set at 250 °C. A calibration curve was established for each standard compound, by injecting a dilution series, prepared in 50% (*v*/*v*) aqueous ethanol and diluted stepwise in a 1:1 ratio. All samples were tested in triplicate. 

#### 2.4.2. Headspace Solid-Phase Microextraction-Gas Chromatography-Mass Spectrometry

Each Baijiu sample was diluted to 10% (*v*/*v*) ethanol with boiled ultrapure water. NaCl (1.5 g) and diluted Baijiu sample (5 mL) were added to a 20 mL glass headspace vial, which was sealed with a PTFE/silicone septum and a screw cap. For quantitative analysis, internal standard mixture (40 μL; IS1:2-dimethylpropanoic acid, 1197.12 μg·L^−1^; L-Menthol: 680, μg·L^−1^; 2-phenylethyl acetate-d3, 146.32 μg·L^−1^; 2-methoxy-d3-phenol, 307.36 μg·L^−1^) was added to each sample. The sample was stirred at 250 rpm for 5 min at 45 °C. After equilibration, a solid-phase microextraction (SPME) arrow fiber (phase: divinylbenzene/carbon wide range/polydimethylsiloxane; thickness 120 um; length 20 mm) that was purchased from CTC Analytics AG (Zwingen, Switzerland) was exposed to the headspace for 50 min at the same temperature and stirring speed. Subsequently, the fiber was introduced into the GC injector port at 250 °C and was subjected to desorption in spitless mode for 5 min. Each sample was analyzed in triplicate.

Gas chromatography–mass spectrometry (GC–MS) analysis was performed on an Agilent 7890 gas chromatograph equipped with an Agilent 5977B mass spectrometer. Samples were analyzed with a DB-FFAP capillary column (60 m × 0.25 mm i.d.; 0.25 μm film thickness, J&W Scientific, Folsom, CA, USA). GC–MS conditions were as described previously, with some modifications [20]. High purity helium (≥99.999%) at a constant flow rate of 2 mL/min was used as carrier gas. The column oven was held at 50 °C for 2 min, increased at 6 °C/min to 230 °C, then held for 15 min. Mass spectra were recorded in electron impact (EI) mode, at an ionization energy of 70 eV. The transfer line and ion source temperatures were 250 °C and 230 °C, respectively. The mass range was 35–350 *m*/*z*, which was scanned at 4.4 scans/s. Scan mode was used for MS data acquisition and extracted ion chromatograms were used for quantification. The retention index (RI) of each compound was calculated from the retention times of n-alkanes (C7–C30), according to a modified Kovats method. Identification of the volatile compounds was performed based on comparisons of their mass spectra and RI with those of pure standards under the same chromatographic conditions, or with RIs reported in the literature (RIlit). A calibration curve for each standard compound was established with a 1:1 dilution series, in aqueous ethanol (10%, *v*/*v*). All samples were tested in triplicate.

### 2.5. Statistical Analysis

A two-way analysis of variance (ANOVA) was performed with rstudio software (version 1, 4.1717, RStudio PBC, Boston, MA, USA) to assess the quality of the sensory data of 18 Baijiu samples obtained from descriptive sensory analysis (the main and interacting effects include sample, panelist, replication, sample × panelist, sample × replication and panelist × replication) by using the summary aov function. A one-way analysis of variance (ANOVA) followed by Duncan multiple comparisons was performed using SPSS version 22.0 for Windows (SPSS Inc., Chicago, IL, USA), to assess differences in the sweetness intensities of the 18 Baijiu samples and in the sweetness intensities of 7 different compositions and compound concentrations in the 4 distilled fractions. A heat map was created with TBtools [21] (V1.09854, South China Agricultural University, Guangzhou, China), of the distribution of compounds in the four fractions. Pearson’s correlation coefficient’s (PCC) were calculated using SPSS version 22.0, to assess the correlation of the 43 compound concentrations with the sensory scores of the 18 Baijiu samples. The correlations between sweetness intensity and compound concentrations (all relative concentrations of compounds were normalized by SPSS version 22.0) were visualized using the superheat function from the superheat package in rstudio software [22]. The significance of the sweetness difference between the two samples of addition and omission tests Section 2.3.4 at probability levels of 5% and 1% was determined according to a predetermined table (two-tailed) in the ISO standard [23]. If the number of common responses was equal to, or greater than the number provided in the table, there was considered to be a sweetness difference between the two samples.

## 3. Results

### 3.1. Evaluation of Sweetness Intensity of Baijiu Samples by Descriptive Sensory Analysis

The sweetness of eighteen Baijiu samples, which were diluted with ultrapure water to the same ethanol content (50%, *v*/*v*), was rated using a scale from 0 (none) to 15 (strong) by 12 well-trained panelists. Each sample was evaluated 48 times, independently (12 panelists over 4 replications). The quality of the sensory data was evaluated by two-way analysis of variance (ANOVA), and it showed that there were significant main effects for sample, panelist and sample × panelist interaction (Appendix A). The significant product effect indicated that the 18 Baijiu samples were perceived as distinguishable based on sweetness. The significant panelist, and sample × panelist interaction effects, were not unexpected in sensory analysis, as individual panelists interpreted stimuli differently [24]. No effects were observed for replication, panelist × replication, or sample x replication. Thus, the sensory data could be considered reliable.

The sweetness intensities of the 18 Baijiu samples were obtained by calculating the average value of 48 sweetness intensity scores (Figure 1); the sweetness intensities of 12 of the samples were from 7 to 12, whereas those of the other six were less than 7. S1 was the sweetest at 11.7, which was significantly (*p* < 0.05) higher than the second highest, S5, so S1 was selected for the subsequent sensory-guided isolation, to identify the sweet flavor components.

### 3.2. Sweetness Screening of Baijiu Fractions by Sensory-Guided Strategy

To identify the Baijiu components that contributed to the sweetness, sample S1 was first separated into volatile and nonvolatile fractions by distillation and these were evaluated by descriptive sensory analysis. The sweetness intensity of the volatile fraction (11.2) was much higher than that of the nonvolatile fraction (1.3), and very close to that of sample S1 (11.7) (Table 1), indicating that the sweet compounds were predominantly volatile. In addition, the sweetness intensity of the volatile fraction was much higher than that of 50% (*v*/*v*) aqueous ethanol (3.0), confirming that there are other volatile compounds contributing to the Baijiu sweetness besides ethanol. Previous studies [8,25] mainly focused on the contribution of non-volatile compounds to food sweetness perception; however, the above findings demonstrated that volatile compounds made a much larger contribution to sweetness perception in Baijiu.

To isolate and characterize the main Baijiu sweetness components, the volatile fraction of S1 was separated by fractional distillation into four fractions (A–D), with A being the most volatile and D the least. After adjusting their ethanol content to 50% (*v*/*v*), these fractions were evaluated by descriptive sensory analysis. The sweet components were mainly present in the more volatile fractions with relatively low boiling points (Table 1), because the sweetness intensity of fractions A (9.5) and B (9.3) was much higher (*p* < 0.05) than that of fractions C (2.4) and D (2.2).

### 3.3. Identification of Potential Sweetness Compounds by HS-SPME Arrow Coupled with GC–MS

To characterize the type and content of aroma compounds GC–MS, coupled with head-space analysis was used for qualitative and semi-quantitative analysis of aroma compounds in the distilled fractions. A total of 88 volatile compounds (Appendix A), including 47 esters, 16 acids, 9 alcohols, 8 aldehydes and ketones, 3 furans, 3 phenols, and 2 terpenes were identified in the distilled frictions; fractions A and B were predominantly composed of esters, aldehydes, and ketones, whereas fractions C and D were predominantly composed of carboxylic acids.

To clarify the compositional differences between the distilled fractions, their compositions were analyzed by cluster analysis and visualized with a heatmap (Figure 2). This confirmed that fractions A and B contained similar compound classes, so they were designated as Group I, whereas fractions C and D, were similar to each other, but distinct from A and B (designated as Group II). Comparison of the compositions of the two groups (Appendix A and Figure 2), showed that the content of the 43 identified compounds in Group I was significantly higher than in Group II (*p* < 0.05), i.e., 32 esters, 7 aldehydes and ketones, 2 alcohols, and 2 acids. Given that the sweetness intensities of fractions A and B were significantly higher than those of fractions C and D, it appears likely that these 43 aroma compounds were primarily responsible for the sweetness of Baijiu.

### 3.4. Screening of Key Compounds Contributing to Baijiu Sweetness by Multivariate Data Analysis (MVDA) 

To determine their relative contributions to the sweetness perception of Baijiu, the 43 aroma compounds were relatively quantified in the 18 original Baijiu samples (Appendix A). The relationship between the relative content of the 43 compounds and the sweetness of the 18 Baijiu samples was then analyzed by their Pearson’s correlation coefficient (PCC) and cluster analysis, then visualized with a heat-map (Figure 3). Baijiu samples with different sweetness levels were clearly distinguished by their aroma compound profiles, and 36 of the 43 aroma compounds were positively correlated with sweetness. Of these 36 compounds, 14 had a significant positive correlation with the sweetness of Baijiu (PCC > 0.5), including 13 esters and one aldehyde. Esters are the primary source of “fruity” and “floral” flavors in food and are also associated with sweetness [26,27,28]; interestingly, the aldehyde, namely octanal has been also described with “fruity” and “citrusy” notes [29]. Fruity aromas are potential candidates for sweetness enhancement and they correlate with sweetness perception [30]. For example, the perceived sweetness of whipped cream samples increased when a strawberry aroma was added, increasing the concentration of a peach aroma increased not only the perceived sweetness intensity of a glucose solution, but also its duration [30] and a medium level of pomegranate aroma significantly increased the intensity of perceived sweetness [31]. However, in the above studies, the influence of sweet aromas on sweetness perception appeared to be olfactory rather than gustatory in origin; a particular aroma did not induce a similar effect when the panelists wore a nose clip and could not smell the aroma [30]. However, in this study, the panelists were still able to perceive sweetness while wearing a nose clip, which suggested that the sweet aroma compounds in Baijiu contributed to sweetness perception by another mechanism. 

### 3.5. Confirmation of the Key Sweetness Compounds by Addition and Omission Tests

To verify the sweetness contributions of the 14 key compounds highlighted by MVDA, the 14 compounds that were positively associated with Baijiu sweetness perception were recombined in 50% (*v*/*v*) aqueous ethanol, at the same concentration as in sample S1 that diluted to 50% ethanol content, to simulate the sweetness of S1 (Table 2). The reconstituted sample (designated MS1) was compared with the original Baijiu sample S1 and 50% (*v*/*v*) aqueous ethanol (control) separately by using the two-alternative forced choice (2-AFC) method. There was no difference in sweetness between MS1 and S1, but the sweetness of sample MS1 was significantly (α = 0.001) higher than 50% (*v*/*v*) aqueous ethanol (Table 3), demonstrating that the 14 compounds were almost entirely responsible for the sweetness of Baijiu.

The sweetness contribution of each individual compound was compared by omission testing (Table 3). Omitting ethyl hexanoate from MS1 produced the greatest reduction in perceived sweetness (*p* < 0.001), followed by omitting hexyl hexanoate or ethyl 3-methylbutanoate (*p* ≤ 0.05); clearly, these compounds made the largest individual contributions to the sweetness of Baijiu. It was notable that the concentration of these three compounds was much higher in Baijiu than in other major distilled liquors [32] and the concentration of ethyl hexanoate was higher in strong aroma-type Baijiu (0.6–2.8 g·L^−1^) than in the other types of Baijiu, which explained why strong-flavor Baijiu was usually the sweetest of the 12 categories [33]. The relatively high concentration of sweetness compounds in strong-flavor Baijiu could be attributed to its production process, because it is fermented in a mud pit, which promotes the formation of ethyl and hexyl hexanoate [33]. Furthermore, none of the 14 aroma compounds had the structural features normally associated with sweet compounds, such as multiple hydroxyl groups (sugars and alditols), chlorine atoms (artificial sweeteners), or α-amino and carboxyl groups (amino acids and small peptides), that can activate sweet taste receptors [9]. Therefore, it is far from clear how these volatile esters contribute to the sweetness of Baijiu; possibilities include that these compounds enhance the response of the sweetness receptor to ethanol or activate sweet taste cell-expressed olfactory receptors. Further research is needed to confirm this.

## 4. Conclusions

This study was the first attempt to evaluate the contribution of volatile compounds to the perception of Baijiu sweetness, by using sensory-guided isolation, assisted by multivariate data analysis. The key compounds contributing to Baijiu sweetness were successfully identified and verified; volatile aroma compounds were the main contributors to Baijiu sweetness, which was successfully reconstituted by a model solution composed of the 14 identified compounds. Omission testing determined that ethyl hexanoate, hexyl hexanoate, and ethyl 3-methylbutanoate made the largest individual contributions to Baijiu sweetness. These findings not only broaden understanding of Baijiu sweetness, but also highlight the major contribution of volatile compounds to sweetness perception and enhance knowledge, which may facilitate future flavor modification of a wide variety of foods and beverages.

## Figures and Tables

**Figure 1 foods-10-02843-f001:**
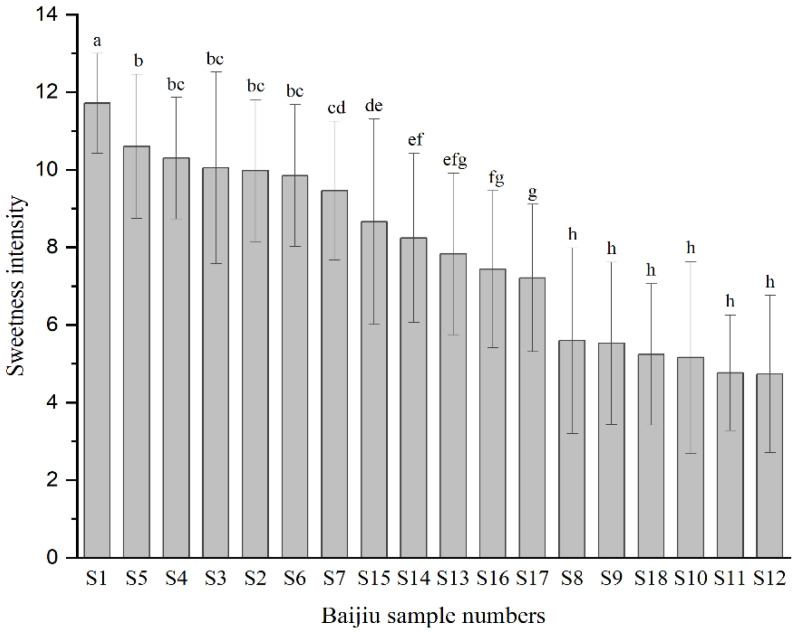
Sweetness intensities of the 18 Baijiu samples. Sweetness intensity was evaluated by 12 sensory panelists using a scale from 0 (none) to 15 (strong). Values are the mean ± standard deviation (SD). The different letters indicate significant differences at *p* < 0.05.

**Figure 2 foods-10-02843-f002:**
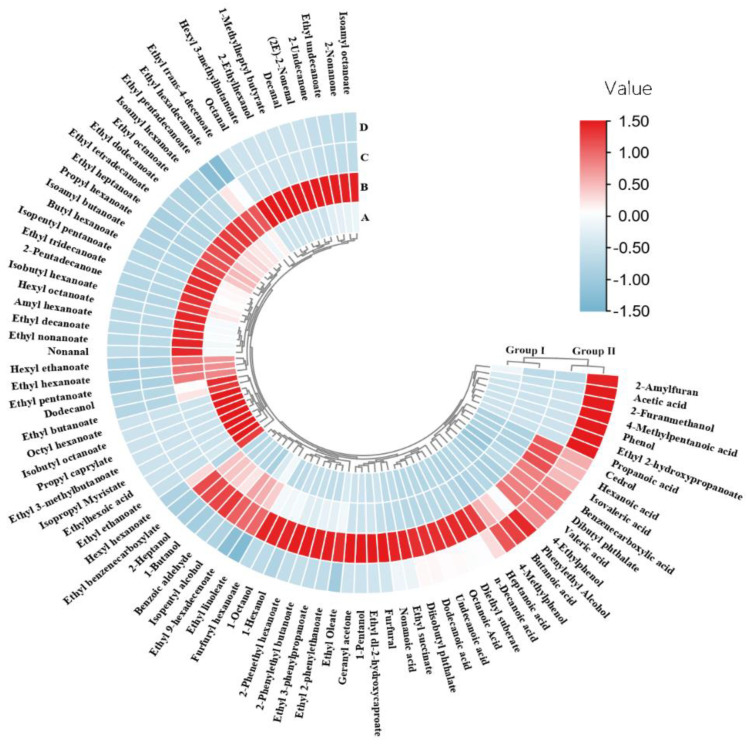
Heat map of aroma compound content of the four fractions distilled from (vacuum condition: A-800 hPa, B-600 hPa, C-400 hPa, D-200 hPa) Baijiu sample S1. Group I and Group II are distinguished by cluster analysis of composition of different fractions: fraction A and fraction B were clustered into Group I, indicating that the two fractions had similar composition (Group II in the same way). Value: standardized value of the concentration of compounds.

**Figure 3 foods-10-02843-f003:**
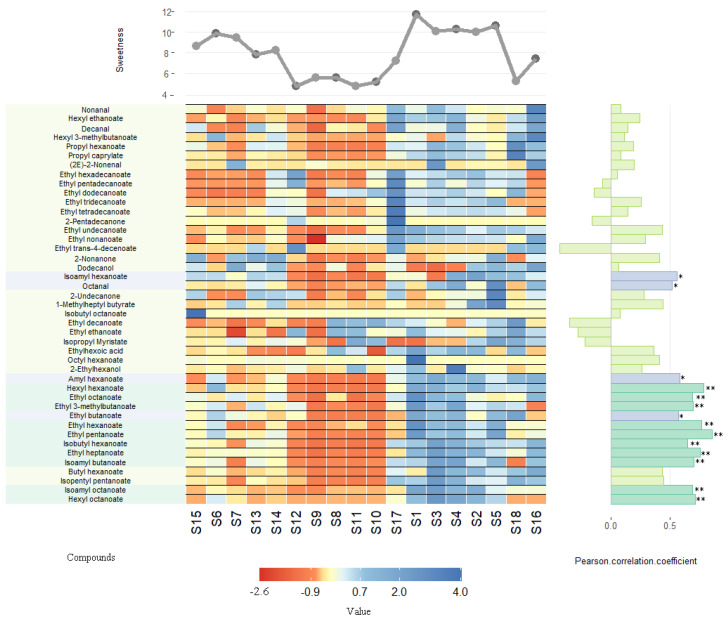
Visualization figure of the correlation between the contents of 43 aroma compounds and sweetness intensity of the 18 Baijiu samples. Compounds: The compounds in Group 1 that are significantly higher concentration (fraction A and fraction B) than in group 2 (fraction C and fraction D) were selected (the data of reactive concentration and visualization figure of these compounds are shown in Appendix A and Figure 2). Sweetness: the sweetness intensities of the 18 Baijiu samples. Value: The value is a normalized value of relative concentration of 43 compounds, which were visualized by heatmap. Pearson correction coefficient (PCC): a value was used to indicate the correlation between the concentration of compounds and sweetness—** highly significant positive correlation (PCC ≥ 0.6), * significant positive correlation (PCC ≥ 0.5).

**Table 1 foods-10-02843-t001:** Sweetness intensity of the fractions prepared from Baijiu sample S1.

Fraction	Original Ethanol Content (%, *v*/*v*)	Vacuum (hPa)	Dilution Factor ^1^	Sweetness Intensity ^2^
Volatile fraction	49	30	1.0	11.2 ± 0.7 ^a^
Nonvolatile fraction	0	30	2.0	1.3 ± 0.3 ^d^
50% aqueous ethanol	50	—	1.0	3.0 ± 0.5 ^c^
Fraction A	62	800	1.2	9.5 ± 0.6 ^b^
Fraction B	74	600	1.5	9.3 ± 0.5 ^b^
Fraction C	43	400	1.1	2.4 ± 0.4 ^c^
Fraction D	0	200	2.0	2.2 ± 0.3 ^c^

^1^ Dilution factor: dilution factor of the different fractions to achieve 50% (*v*/*v*) ethanol content, if the original ethanol of fraction is higher than 50% (*v*/*v*), dilute it with ultrapure water, otherwise, use high purity edible ethanol. ^2^ Values of sweetness intensity are the mean ± SD, the different letters indicate significant differences at *p* < 0.05. ”—”: Components not prepared by vacuum distillation.

**Table 2 foods-10-02843-t002:** Concentration of the 14 potential compounds, which may contribute to sweetness of Baijiu in sample S1 diluted to 50% ethanol content (detailed information of quantitative analysis is shown in the Appendix A).

No.	CAS	RI (FFAP)	Compound	Concentration(mg·L^−1^)
1	539-82-2	1136	Ethyl pentanoate ^1^	59.40 ± 0.00
2	123-66-0	1253	Ethyl hexanoate ^1^	2172.72 ± 0.19
3	1117-55-1	1798	Hexyl octanoate ^2^	4.15 ± 0.13
4	108-64-5	1058	Ethyl 3-methylbutanoate ^2^	16.09 ± 0.85
5	106-30-9	1343	Ethyl heptanoate ^1^	40.01 ± 0.00
6	106-27-4	1291	Isoamyl butanoate ^2^	2.50 ± 0.12
7	6378-65-0	1601	Hexyl hexanoate ^2^	113.58 ± 3.84
8	2035-99-6	1650	Isoamyl octanoate ^2^	2.01 ± 0.02
9	105-79-3	1357	Isobutyl hexanoate ^2^	6.13 ± 0.30
10	105-54-4	1028	Ethyl butanoate ^1^	225.76 ± 0.03
11	2198-61-0	1459	Isoamyl hexanoate ^2^	46.42 ± 2.03
12	106-32-1	1434	Ethyl octanoate ^1^	76.89 ± 0.01
13	124-13-0	1300	Octanal ^2^	1.98 ± 0.13
14	540-07-8	1509	Amyl hexanoate ^2^	10.76 ± 0.33

^1^ These compounds were quantified by GC-FID. ^2^ These compounds were quantified by GC-MS/head-space analysis.

**Table 3 foods-10-02843-t003:** Addition and omission tests via two-alternative forced choice (2-AFC) by 24 panelists.

Test	Pair	*n*/24 ^1^	Significance ^2^
Addition tests			
1	MS1: Sample S1	12/24	ns
2	MS1:50% alcohol aqueous solution	22/24	***
Omission tests			
1	MS1: omit ethyl hexanoate	23/24	***
2	MS1: omit hexyl hexanoate	18/24	*
3	MS1: omit ethyl 3-methylbutanoate	18/24	*
4	MS1: omit isoamyl butanoate	16/24	ns
5	MS1: omit isoamyl octanoate	16/24	ns
6	MS1: omit ethyl pentanoate	11/24	ns
7	MS1: omit ethyl heptanoate	12/24	ns
8	MS1: omit hexyl octanoate	15/24	ns
9	MS1: omit ethyl octanoate	13/24	ns
10	MS1: omit isobutyl hexanoate	11/24	ns
11	MS1: omit amyl hexanoate	13/24	ns
12	MS1: omit ethyl butanoate	11/24	ns
13	MS1: omit isoamyl hexanoate	14/24	ns
14	MS1: omit octanal	14/24	ns

^1^ The number of panelists choosing MS1 as the sweeter sample. ^2^ Significance: “***”, *p* < 0.001 (very highly significant); “*”, *p* < 0.05 (significant); “ns”, *p* > 0.05 (not significant).

## Data Availability

The data supporting the findings of this study are available from the corresponding author upon request.

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
