# Peer review of "Exploring the Mystery of the Sweetness of Baijiu by Sensory Evaluation, Compositional Analysis and Multivariate Data Analysis"

_foods, 2021, doi:10.3390/foods10112843_

Round 1

Reviewer 1 Report

The authors tested a range of Baijiu samples for perceived sweetness using a sensory panel and further fractionated the samples that had the highest perceived sweetness to better understand which compounds drive this attribute. They found that the volatile fraction, particularly ethyl hexanoate, is the main contributor in sweetness as opposed to the non-volatile fraction. I believe that this is a novel and relevant finding for the research community. 

line 57: need to double check that multivariate data analysis or multivariate statistical analysis is consistent throughout

Materials and Methods:

general comment: is there any additional basic chemistry on the samples that can be included (e.g. titratable acidity, pH, residual sugar) for table S1? might not be necessary but would be nice to have since those factors are known to impact sweetness. 

lines 140-142: can you include dilution factors of the different fractions to achieve 50% ABV? how important is the dilution factor on the perception of sweetness?

section 2.3.4. (line 164): are the MS1 compound concentrations for the selected 14 compounds the same as shown in table 2?  The dilution factor makes it ambiguous as to whether the concentrations were matched at 50% ABV at the same levels found at the original baijiu (69% ABV) or after it was diluted to 50% ABV (~1.4x dilution factor). I think this is touched on starting line 345 but it is somewhat confusing in the materials and methods. 

line 209: what are the SPME Arrow dimensions and material?

line 364-366: since this is unpublished work, I believe it should be removed.

Figure 3: I like the visualization overall but the "Value" parameter which goes from pale blue to purple is a bit ambiguous. Is this a normalized value of relative concentration of these compounds?

Table 2: I think the table demonstrates analytical rigor of the quantitative analysis, but for simplicity it might be better to only include the columns for the 'compound' and its corresponding 'concentration'. The remaining analytical information might be a better fit for supplementary data.

Reviewer 2 Report

The paper is well-written and follows a great structure.

More information is needed on the sensory panellist, what were their previous experience? Are they trained? How was the reference samples made?

Perhaps the authors should create a flow chart or some sort to further visualise their tests and the samples that they have evaluated.

Please remove the stats approach in Section 2.3.4 and merge it together with Statistical Analysis section.

What was included as main and interaction effects? PLease elaborate. AVOVA to ANOVA?

No post-hoc comparisons?

Ensure all software have their manufacturer, and city.

More explanation is required for MVDA especially using the PCC under statistical analysis section, how does one recreate the heat map and it seems some significance testing was also conducted.

Figure 3. PPC to PCC?
